# Replicating and Improving GAN2Shape Through Novel Shape Priors and Training Steps

## Reproducibility Summary

**Scope of Reproducibility**

Pan et al. [2021] propose an unsupervised method named GAN2Shape that purportedly is able to recover 3D information stored in the weights of a pre-trained StyleGAN2 model, to produce 3D shapes from 2D images. We aim to reproduce the 3D shape recovery and identify its strengths and weaknesses.

**Methodology**

We re-implement the method proposed by Pan et al. [2021] with regards to 3D shape reconstruction, and extend their work. Our extensions include novel prior shapes and two new training techniques[1] While the code-base relating to GAN2Shape was largely rewritten, many external dependencies, which the original authors relied on, had to be imported[2]. The project used 189 GPU hours in total, mostly using a single Nvidia K80, T4 or P100 GPU, and a negligible number of runs on a Nvidia V100 GPU.

**Results**

We replicate the results of Pan et al. [2021] on a subset of the LSUN Cat, LSUN Car and CelebA datasets and observe varying degrees of success. We perform several experiments and illustrate the successes and shortcomings of the method. Our novel shape priors improve the 3D shape recovery in certain cases where the original shape prior was unsuitable. Our generalized training approach shows initial promise, but has to be confirmed with increased computational resources.

**What was easy**

The original code is easily runnable on the correct machine type (Linux operating system and CUDA 9.2 compatible GPU) for the specific datasets used by the authors.

**What was difficult**

Porting the model to a new dataset, problem setting or a different machine type is far from trivial. The poor cohesion of the original code makes interpretation very difficult, and that is why we took care to re-implement many parts of the code using the decoupling principle. The code depends on many external implementations which had to be made runnable, which caused a significant development bottleneck as we developed on Windows machines (contrary to the authors). The exact loss functions and the number of training steps were not properly reported in the original paper, which meant it had to be deduced from their code. Certain calculations required advanced knowledge of light-transport theory, which had no familiarity to us, and had to be mimicked and could not be verified.

**Communication with original authors**

We did not communicate with the original authors.

---

[1]Our code is available at `https://anonymous.4open.science/r/GAN-2D-to-3D-03EF`.
[2]All depencies are declared in section 3

Submitted to ML Reproducibility Challenge 2020. Do not distribute.

# 1 Introduction

Image generation has been a hot topic within generative models as they represent an intuitive problem whose results are easily accessible by the public. One of the models that has received a lot of public attention is StyleGAN (Karras et al. [2019]). The network's architecture has been refined through multiple iterations in StyleGAN2 (Karras et al. [2020b]), StyleGAN2-ADA (Karras et al. [2020a]) and StyleGAN3 (Karras et al. [2021]). StyleGAN2 improves on the first version by, among other things, adding a projection method onto the latent space, which allows the inversion an image into its latent representation.

Methods like GAN2Shape (Pan et al. [2021]) aim at exploiting the information that is already stored in the generator of a pre-trained StyleGAN2 model to go beyond generating synthetic 2D images. In particular, this method aims to extract the 3D shape of the preeminent object in any image. This is intuitively possible due to the size of the training dataset of the StyleGAN2 model, and its ability to generate images of an object from multiple views and lighting directions by varying $\mathbf{w}$. The authors of GAN2Shape use StyleGAN2 networks pre-trained on different dataset categories and five different feature extraction models to derive the shape information for images belonging to the same dataset categories. This method, compared to many others (Lunz et al. [2020], Henzler et al. [2019], Wu et al. [2015], Wang et al. [2019]), has the advantage of being completely unsupervised, and not requiring a change in the training process of the classical 2D GAN.

In this article, we describe our replication of GAN2Shape (Pan et al. [2021]) and report mixed results. We perform several experiments and we illustrate the successes and shortcomings of the method. Further, we extend the method improving the original results in several cases.

# 2 Scope of reproducibility

The authors of GAN2Shape make the following claims:

1. Their framework does not require any kind of annotation, keypoints or assumption about the images

2. Their framework recovers 3D shape with high precision on human faces, cats, cars, buildings, etc.

3. GAN2Shape utilizes the intrinsic knowledge of 2D GANs

4. The 3D shape generated immediately allows for re-lighting and rotation of the image.

# 3 Methodology

Our initial intent of re-implementing the source code from from the description of the paper had to be abandoned due to lack of detailed information of some key points in the method. We, therefore, decided to follow a different approach integrating both the details from the authors' code and the paper's description. While trying to always base our implementation on the paper's description we found some parts (particularly, the loss functions) that differed from the actual code and decided to follow the latter instead.

The resources we used were mainly the authors' code, the code and documentation of all the out-sourced methods the authors borrowed: StyleGAN2 Karras et al. [2020b] (code), Unsup3D Wu et al. [2020] (code), Semseg Zhao [2019] (code) and BiSeNet Yu et al. [2018, 2021] (code). The GPUs used were multiple and varied depending on availability: Nvidia Tesla K80, T4, V100, P100.

## 3.1 Model descriptions

To extract the implicit 3D knowledge of pre-trained StyleGAN network, Pan et al. [2021] propose an elaborate scheme involving five different neural networks. Each network models a particular quantity corresponding to the view and lighting directions, the depth of the image, and the albedo. The **View** and **Light** ($V$ and $L$, resp.) networks operate in a *encoder* type manner, trying to obtain a low-dimensional vector representation of the camera view direction $\mathbf{v}$ and the direction of light $\mathbf{l}$ illuminating the object in the picture. The **Depth** and **Albedo** ($D$ and $A$, resp.) networks utilize *auto-encoder* architectures[3] to obtain image-resolution depth maps $\mathbf{d}$ and diffuse reflections (albedo) $\mathbf{a}$ off the object's presumed surface.

The real GAN knowledge extraction happens in the final network, the **Offset** encoder $E$, combined with the pre-trained StyleGAN2 generator, $G$. The offset encoder aims to learn a latent representation $\mathbf{w}$ of images with randomly sampled

---

[3]We refer to tables 5-7 of the original paper (Pan et al. [2021]) for the exact architectures.

76 view and light directions, *pseudo-samples*. Paired with $G$, this allows the creation of new realistic samples $\tilde{\mathbf{I}}_i = G(\mathbf{w}_i^{'})$
77 with new view and lighting directions, denoted *projected samples*. The projected samples then serve as extended
78 training data, providing multiple view-light direction variations of the original image.

79 To use the components $\mathbf{v}$, $\mathbf{l}$, $\mathbf{d}$ and $\mathbf{a}$ to obtain a reconstructed image, the authors utilize a pretrained neural renderer
80 developed by Kato et al. [2017], which we denote by $\Phi$.

### 3.1.1 Training Procedure

82 The training process of this method can be divided into 3 different steps, where the different networks involved are
83 trained separately. In the original paper, these steps are done sequentially and for one image at a time, as shown in
84 Figure 1, and each step is repeated multiple times before moving into the following one. The result is a model that can
85 predict the depth map for only one image. All of the networks are trained using the Adam optimization algorithm.

86 **Prior pretraining.** Before attempting to learn the true shape of an object, the depth network is initialized by pretraining
87 it on a fixed prior shape. For this purpose Pan et al. [2021] propose to use an *ellipsoid* shape as the shape prior. We
88 utilized this ellipsoid prior to reproduce the results of Pan et al. [2021], and we extended their work by also evaluating
89 two new different priors.

90 **Step 1** optimizes only the $A$ network according to Equation 1. Given an input $\mathbf{I}$, the first four networks predict their
91 components $\mathbf{v}$, $\mathbf{l}$, $\mathbf{d}$, $\mathbf{a}$, and we obtain a reconstructed image $\hat{\mathbf{I}} = \Phi(\mathbf{v}, \mathbf{l}, \mathbf{d}, \mathbf{a})$. [4]

$$\mathcal{L}_{\text{step1}}(\mathbf{I}, \hat{\mathbf{I}}) = \|\mathbf{I} - \hat{\mathbf{I}}\|_1 + \lambda_s \mathcal{L}_s(D(\mathbf{I})) + \lambda_p \mathcal{L}_p(\mathbf{I}, \hat{\mathbf{I}}) \tag{1}$$

92 **Step 2** optimizes the $E$ network according to Equation 2. Using the $\mathbf{d}$ and $\mathbf{a}$ components given in the last step 1
93 iteration, and random directions $\mathbf{v}_i^{'}$, $\mathbf{l}_i^{'}$, we generate $N_p$ new pseudo-images $\mathbf{I}_i^{'}$. For each $\mathbf{I}_i^{'}$ we predict $\Delta \mathbf{w_i} = E(\mathbf{I}_i^{'})$,
94 which serves as input to the StyleGAN generator network $G$ and obtain the projected images $\tilde{\mathbf{I}}_\mathbf{i}$.

$$\mathcal{L}_{\text{step2}}(\mathbf{I}) = \frac{1}{N_p} \sum_{i=1}^{N_p} \|\mathbf{I}_\mathbf{i}^{'} - G(\mathbf{w} + E(\mathbf{I}_\mathbf{i}^{'}))\|_1 + \lambda_1 \|E(\mathbf{I}_\mathbf{i}^{'})\|_2 \tag{2}$$

95 **Step 3** optimizes the $L$, $V$, $D$ and $A$ networks according to Equation 3. It consists in part of $\mathcal{L}_{\text{step1}}$. The second part
96 utilizes the projected samples from the last iteration of step 2. For each projected sample $\tilde{\mathbf{v}}_\mathbf{i} = V(\tilde{\mathbf{I}}_\mathbf{i})$, $\tilde{\mathbf{l}}_\mathbf{i} = L(\tilde{\mathbf{I}}_\mathbf{i})$ is
97 calculated. Combined with $\mathbf{d}$ and $\mathbf{a}$ from the original image, they can be used to reconstruct each projected sample
98 from the components $\bar{\mathbf{I}} = \Phi(\tilde{\mathbf{v}}_i, \tilde{\mathbf{l}}_i, \mathbf{d}, \mathbf{a})$.

$$\mathcal{L}_{\text{step3}}(\mathbf{I}, \bar{\mathbf{I}}) = \frac{1}{N_p} \sum_{i=1}^{N_p} [\mathcal{L}_p(\mathbf{I}, \bar{\mathbf{I}}_i) + \|\mathbf{I} - \bar{\mathbf{I}}_i\|_1] + \mathcal{L}_{step1}(\mathbf{I}, \hat{\mathbf{I}}) + \lambda_2 \mathcal{L}_s(\mathbf{D}(\mathbf{I})) \tag{3}$$

99 **Stages.** The steps are repeated for a number of *stages*. In each, the steps are trained for a different number of iterations
100 (see Table 1 in Appendix A for details).

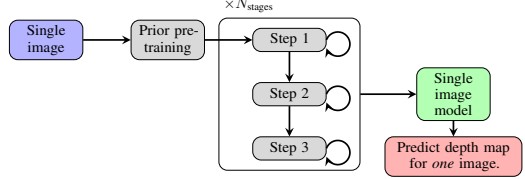

Figure 1: Schematic of the original training process.

---

[4]$\mathcal{L}_p$ is a neural network trained to predict similarities between images (Johnson et al. [2016]) and $\mathcal{L}_s$ is a term that encourages smoothness of the resulting depth maps (as described in Zhou et al. [2017]). We refer to our code for the weights $\lambda_i$.

### 3.1.2 Novel Shape Priors

The first novel prior we consider is a masked box. Using the mask returned by the parsing model developed by Zhao et al. [2017] we extrude the relevant object from the background, in a step-like manner. Improving on this idea, we also smooth the transition from the object to the background. This is done by using three 2D convolutions, where we convolve the masked box shape with a $11 \times 11$ filter of ones. Renormalizing the convolved shape, we obtain Figure 2c denoted as 'smoothed box'.

The last prior we tested is obtained by normalizing the score (or "confidence") that the parsing model gives to each pixel. We use this confidence to project the object, i.e. a pixel that is within the category with more confidence will be farther projected. This prior is similarly smoothed by convolutions and is denoted as 'confidence based'.

Figure 2 shows a visual representation of the prior shapes used for an example image taken from the Celeba dataset.

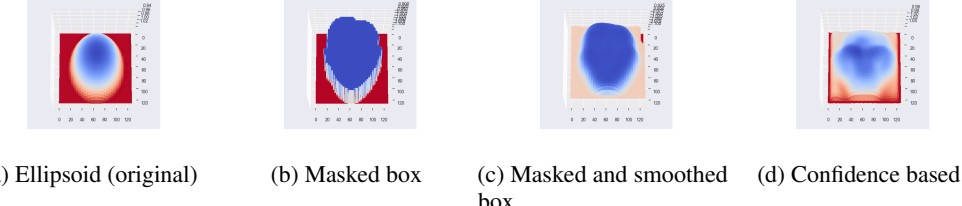

(a) Ellipsoid (original)    (b) Masked box    (c) Masked and smoothed box    (d) Confidence based

Figure 2: Original vs. our novel shape priors, shown on the Celeba (face) dataset

### 3.2 Generalized Training Procedure

Motivated by our findings on the forgetting of previously seen images, extensively explained in section 5.3.2 and the appendix A.5, we propose an alternative training procedure to favor a general model $M^*$ usable for all images belonging to the same distribution as the training dataset $\mathcal{D}$. We propose to pretrain the depth net $D$ on all images first, instead of repeating the process for each image. We also modify Step 1, 2 and 3 by greatly lessening the number of iterations given to a single image and breaking up the sequential training of the original method into a few iterations per example, and instead introducing $N_e$ *epochs* and batch training to compensate, increase resource utilization and training speed.

To facilitate understanding of our modifications to the training procedure, we provide a schematic in Figure 3. It can be compared to the original shown in Figure 1.

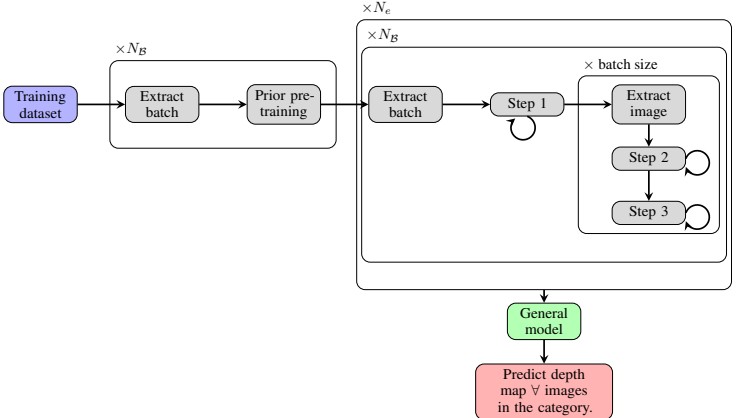

Figure 3: Schematic of our new training process designed to favor generalization.

### 3.3 Datasets

We aimed to reproduce the authors' results on the LSUN Car, LSUN Cat (Yu et al. [2015]) and Celeba (Liu et al. [2015]). From these datasets, the authors selected a subset consisting of 10 images of cars, 216 images of cat faces, and 399 celebrity faces. Like the authors, we used RGB images of three color channels, resized to $128 \times 128$ pixel resolution. No further preprocessing was applied.

### 3.4 Hyperparameters

For replication purposes, the original hyperparameters by Pan et al. [2021] were used, but we also tried tuning some parameters that we believe are key to the method: the number of projected samples, $N_p$, for each image and the number of epochs for pre-training the depth network. $N_p$ was varied within $\{2, 4, 8, 16, 32\}$. In our tests we found the values 4, 8 and 8, respectively for the LSUN Car, LSUN Cat and Celeba dataset, to be the threshold after which the improvements in image quality start greatly decreasing (see subsection A.8 in Appendix for more details).

The number of epochs for the depth network pretraining was varied within $\{100, 500, 1000, 2000\}$. This pretraining affects how irregular the depth map predictions are. We believe that using a threshold for the loss to check the convergence would be preferable as the number of epochs selected by the authors (1000) is enough in most cases but not in all. We attribute irregularity in some of our results to this issue.

### 3.5 Experimental setup and code

For each dataset we run our implementation of the framework from Pan et al. [2021] on the images that were selected by the authors, the procedure saves a checkpoint for each network. These checkpoints are later fed the original image to get the generated result. The evaluation of the results was only qualitative as all the datasets we explored do not have a ground truth for comparison. We instead relied on a manual evaluation.

Our code is available at `https://anonymous.4open.science/r/GAN-2D-to-3D-03EF`. Our results are available interactively under the docs folder.

### 3.6 Computational requirements

Most of the experiments we ran were on a Intel(R) Xeon(R) CPU @ 2.20GHz with 2 cores available and a Nvidia Tesla P100-PCIE-16GB. Since the framework described by Pan et al. [2021] is instance-specific, we report the average time for completing the projection of a single image: 96m and 28s for an image in the Celeba dataset, 95m and 43s for a LSUN Cat image and 74m and 32s for a LSUN Car image.

## 4 Results

The model correctly learned the shape and the texture of many images, while some examples were less successful than others. For example, the model converged to believable shapes for two of the cars in Figure 4, but the shape of the right-most car is debatable.

In the following sections we show the reconstructed depth map and 3D projection of some images chosen as representative of the dataset. All of the images that follow have the background cut from the actual object, this was only done for ease of illustration and was not done for the actual training process since the original authors do not mask the background in all cases. It is also difficult to illustrate the results fairly in 2D images, so we invite the reader to visit our website with *interactive* 3D plots [5].

### 4.1 Results reproducing original paper

#### 4.1.1 LSUN Car

We present the results on LSUN Car dataset in Figure 4. Most features are projected in the right direction and show details that are correctly outward projected from the main object. This result supports all the claims made in section 2 as we did not use any annotation or assumption for the images, many details were retrieved with high precision using the StyleGAN knowledge and we were able to easily make a rotation of the image (see interactive web-page).

#### 4.1.2 LSUN Cat

The second experiment was executed on the LSUN Cat dataset. The results are a slightly poorer compared to the the LSUN Car dataset. The face of the cats gets properly recognized, but some details like the nose are not protruded from the rest of the face and are generally on the same plane, see Figure 4. Some images present some irregularities in the form of spikes and hills (d). The rotation (f) does not result in a completely natural image as part of the face of the cat appears on the same plane. This experiment does not support claims 2 and 4 in some cases (e.g. figures 4 (d) and (f) negate claims 2 and 4 respectively) while it does for claims 1 and 3 (section 2).

Additional results such as for the Celeba dataset, can be found in the Appendix A.

---

[5]Due to the anonymization of the report, we instead refer to the html files under the docs folder in our code

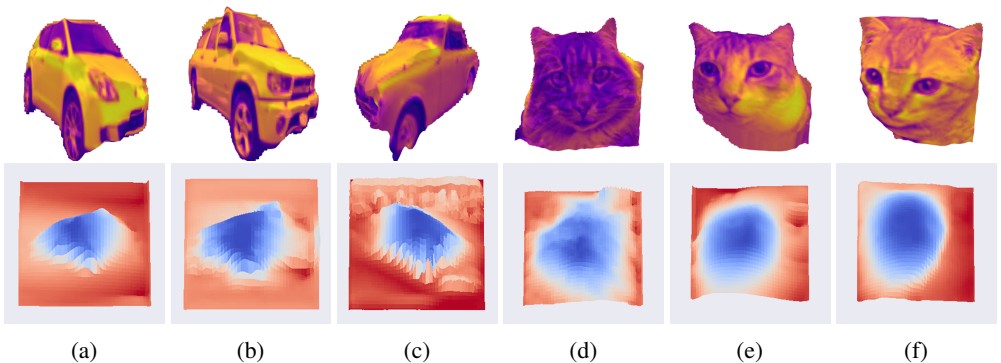

(a)   (b)   (c)   (d)   (e)   (f)

Figure 4: LSUN Car and Cat

## 4.2 Results beyond the original paper

### 4.2.1 The effects of shape priors

**No prior.** To confirm our suspicions that this method would not work at all without a shape prior, briefly mentioned in 3.1.1, we ran a test on one image from the LSUN Car dataset without any prior pre-training, and with random initialization. The reconstruction objective is still satisfied very well, but it has converged to an extremely noisy depth map (see Figure 8 in Appendix A). It shows that this method would not work without a strong shape prior to guide it towards a reasonable shape.

**Smoothed Box Prior.** The first experiment was done by testing the first of the prior shapes presented, the smoothed box prior. Figure 5 shows the smoothed box prior tested on the LSUN Cat and Celeba dataset where it can be seen how it is better at understanding the structure of the nose and face in general (see Appendix A for more details).

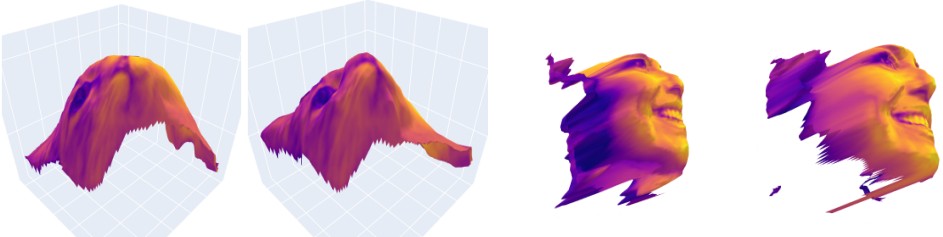

Figure 5: Example result for two different image examples from the LSUN Cat and Celeba datasets. For each example, the left-most figure corresponds to the ellipsoid and right-most figure corresponds to the smoothed masked box prior.

### 4.2.2 Generalized Training Procedure

We demonstrate the results of our new training loop on LSUN Cat. We note again that the difference to the previous demonstration on LSUN Cat, is that a single network $D^*$ was used to predict all of the images, as opposed to a different network $D_i$ for each image $\mathbf{I}_i$. The general model was trained on a limited subset of 30 images from LSUN Cat. It was trained for a modest 60 epochs which results in approximately 60% of the weight updates per image of the original method. Figure 6 shows the projection of some images from the LSUN Cat dataset. One can observe that the method recognizes the general structure of the cat's face but also presents some artefacts in some specific parts of the face e.g. the second cat's cheek is further projected than where it should and similarly for the third cat's chin.

### 4.2.3 Improved initialization

Our final experiment is inspired by the observations reported in sections 5.3.1 and 3.4. We experiment with drastically increasing the number of pseudo-samples $N_p$ from 16 to 128 for 10 short epochs, in which each training step is performed only once. We observe see marginal improvement in the predicted shape (Figure 6) and larger improves in the smaller details/features. See the appendix A.7 for further detail.

Training step 1 was not changed and it is allowed to converge in the first stage, as it does not involve the projected samples. See Table 2 in the appendix for an exact description of the number of iterations. All other parameters were left

as in subsubsection 4.2.1, with the smoothed box prior. We experimented with two of the worst performers from the LSUN Cat dataset to evaluate whether this method could improve the results, see Figure 16. We applied the same idea to the general model described in sections 3.2, 4.2.2 and saw improvements, see Figure 6. The results can be compared to Figure 14.

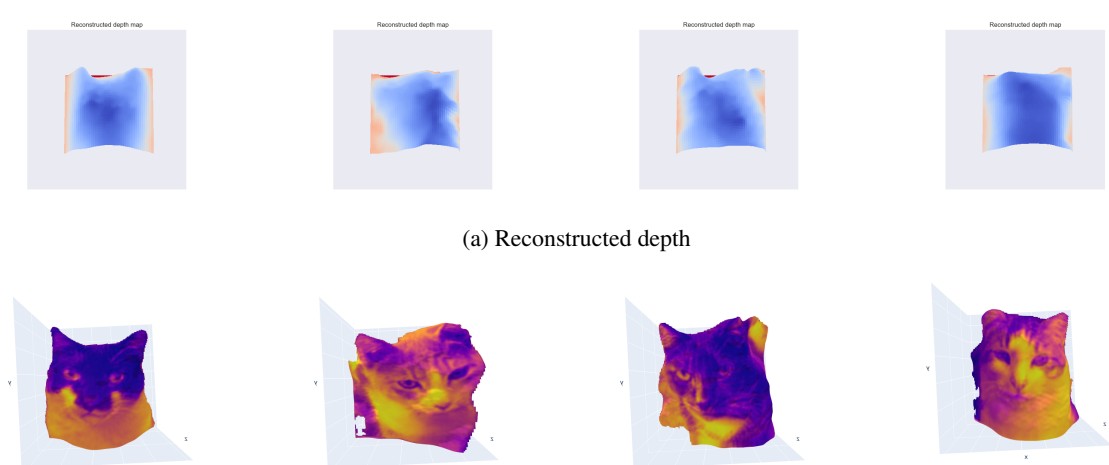

(a) Reconstructed depth

(b) Reconstruced 3D image

Figure 6: Depth map predictions for a few image samples from the training set $\mathcal{D} \subset$ LSUN Cat dataset, all using one and the same **general** model $M^*$ trained with **initialization iterations.**

# 5 Discussion

## 5.1 What was easy

The authors provide a clear specification of the Python package dependencies, as well as other dependencies. Additionally, they provide scripts for easy downloading of a select few datasets and pre-trained model weights. They precisely state how to execute the training script and how to run the model for evaluation. Note that this refers to running the original code and that modifying and extending the code brought many difficulties, as explained in the next section.

## 5.2 What was difficult

The paper by Pan et al. [2021] did not contain enough information for a successful reimplementation. Many details had to be discerned or guessed from their code. Furthermore, the quality of said code does not allow for a quick interpretation. For example, deducing the training loop and the number of iterations for each step was further complicated by the poor cohesion of the original code: the trainer script was heavily mingled with model class, using class members of the model object to increment training steps and nested function calls back and forth between the trainer and model classes.

The components $\mathbf{v}$, $\mathbf{l}$, $\mathbf{d}$ and $\mathbf{a}$ were not enough to pass in to the neural renderer to reconstruct an image. In reality, several calculations of quantities such as diffuse shading and texture were needed to be fed into the neural renderer, using concepts from light transport theory that were not mentioned in the paper.

Another difficulty was the heavy reliance on external pre-trained neural networks. The neural renderer Kato et al. [2017], in particular, posed several problems. The major one was incompatibility with Windows machines. To be able to develop on our personal machines, we had to make manual edits of the neural renderer script and different CUDA files.

Another challenge with this method is the lack of objective quantitative metrics to evaluate the success of the models. One instead has to rely almost entirely on qualitatively gauging the shape reconstructions by eye.

### 5.3 Conclusions

#### 5.3.1 Variability of the results

We observed that the method is very sensitive to various random factors and identical runs may yield different results, see Figure 12. One factor may be the random initialization of the networks, but we do not believe it is the dominating factor, since the depth network is pre-trained on a fixed prior shape each run. Rather, as mentioned by the authors Pan et al. [2021], the quality of the projected samples varies. Additionally, we only sample $8 - 16$ different view-light directions in each step 2 iteration, which may be too few projected samples for a robust model. Since this sampling is random, increasing the number of samples should assure the inclusion of meaningful view and light projections (experimental backing in the Appendix A).

#### 5.3.2 Catastrophic forgetting

We have observed that the instance-specific model forgets the previous training images (see Appendix A.5, Figure 13), and thus has no generalization capability. This is not necessarily a problem if one has time and computational resources. It can also be argued that this is exactly what is intended with this model, and that generalization is up to the training dataset of the StyleGAN model. It does, however, limit the usefulness of the model. As an example, the training time for one $128 \times 128$ pixel RGB image using a Tesla K80 GPU was about 2.5 hours, which seems exceedingly costly for just one low-resolution depth map. We argue that a *general* model would have more use. The ideal scenario would be a model $D^*$ trained on $\mathcal{D}$ that is able to accurately predict $\mathbf{d}_i = D^*(\mathbf{I}_i) \, \forall \, \mathbf{I}_i \in \mathcal{D}$, and even extend to unseen testing data belonging to the same distribution as $\mathcal{D}$. This discussion is what urged us to explore the altered training procedure of sections 3.2 and 4.2.2.

#### 5.3.3 Final conclusions

We were able to replicate some of the results of Pan et al. [2021] on the datasets LSUN Car, LSUN Cat and Celeba. We identified several failure modes and limitations of the model, and back it up with experimental evidence. Examples are the variability and sensitivity to the projected samples, the heavy dependence on shape priors and the computational costliness of the single-use model - all of which were not adequately accounted for in the original paper.

We propose a new prior shape, the smoothed box prior, that has shown very promising results especially for fine details and complex object structures. We propose a second prior shape, confidence-based, that has shown best results in the face dataset. We finally suggest two new training procedures that produce better results and are better at generalizing than the original model by Pan et al. [2021].

We recognize the limitations of this work as we were only able (due to the restricted computational power) to test the method on part of the dataset. For example, the Cat's dataset used by the authors contains more than 200 images but we were able to only test few of them. We speculate that some images in the dataset could yield better results than those reported here. However, we believe that few bad projected images should be enough to claim the uneffectiveness of the method at least in some particular cases.

Another limitation of our work is the lack of quantitative evaluation methods. The original authors propose their results also on the BFM benchmark Paysan et al. [2009] where it is possible to use some metrics to accurately evaluate the results.

### 5.4 Future work

We speculate that it would be interesting to adapt the same method to StyleGAN3 (Karras et al. [2021]) where the network has been modified to support training with fewer samples, leaving the question if the network still retains enough information that is needed for GAN2Shape to work. Future work could also explore the use of our priors on datasets where the original method failed (e.g. the LSUN Horse dataset). We speculate that, since our prior captures the boundaries of the object very well (compared to the ellipsoid where the boundaries are only used to position the origin), it could achieve better results in complex 3D objects where the shape cannot be simplified into an ellipse. A limitation of this method is that it does not use voxels, but learns a height map. This disallows realistic shape reconstructions and more complex geometries with multiple x and y values for each z value etc. Future work should investigate whether this model could be extended to predict voxels instead of height maps. Given our promising results with the generalizing trainer, which was obtained through only a few epochs of training, we believe that it should be further explored with increased epochs and training set size.

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

# A  Appendix

## A.1  Hyperparameters

| Stage | Iterations/step |
|---|---|
| 1 | $[700, 700, 600]$ |
| 2, 3, 4 | $[200, 500, 400]$ |

Table 1: Specification of the different stages for the single-image model.

| Stage | Iterations/step | $N_p$ |
|---|---|---|
| 0 | $[700, 0, 0]$ | 16 |
| 1-10 | $[1, 1, 1]$ | 128 |
| 11 | $[1, 700, 600]$ | 16 |
| 12, 13, 14 | $[200, 500, 400]$ | 16 |

Table 2: Specification of the different stages for the single-image model with **initialization iterations**

| Epochs | Iterations/step | $N_p$ |
|---|---|---|
| 60 | $[13, 22, 18]$ | 16 |

Table 3: Specification of the iterations/step for the generalized model.

| Epochs | Iterations/step | $N_p$ |
|---|---|---|
| 10 | $[13, 1, 1]$ | 128 |
| 60 | $[13, 22, 18]$ | 16 |

Table 4: Specification of the iterations/step for the generalized model with **initialization iterations**

Table 5: Hyperparameters for the general model with initialization iterations on the LSUN Cat dataset.

| Parameter | Value |
|---|---|
| n_epochs_prior | 1000 |
| n_epochs_generalized | 70 |
| n_epochs_init | 10 |
| n_init_iterations | 8 |
| batch_size | 10 |
| channel_multiplier | 1 |
| image_size | 128 |
| z_dim | 512 |
| root_path | data/cat |
| learning_rate | 0.0001 |
| view_scale | 1 |
| refinement_iterations | 1 |
| n_proj_samples | 16 |
| rot_center_depth | 1.0 |
| fov | 10 |
| tex_cube_size | 2 |

We refer to our GitHub repository for a complete declaration of all hyperparameters for all datasets `https://anonymous.4open.science/w/GAN-2D-to-3D-03EF`.

## A.2  Additional replication results

### A.2.1  Celeba

The third experiment conducted on the Celeba dataset shows that most of the face are correctly portrayed with the only exception of the border of the face e.g. chin and forehead that sometimes is not included in the projection (see Figure 7

(b)). Also we found out that the method does not behave well with faces that are viewed from the side (see Figure 7 (c)) where the face still gets a projection as it was viewed from the front. As a consequence of this, the rotation of side faces does not result in a good image. This experiment supports claims 1-4 (section 2) only for some faces and claims 1 and 3 for those viewed from the side.

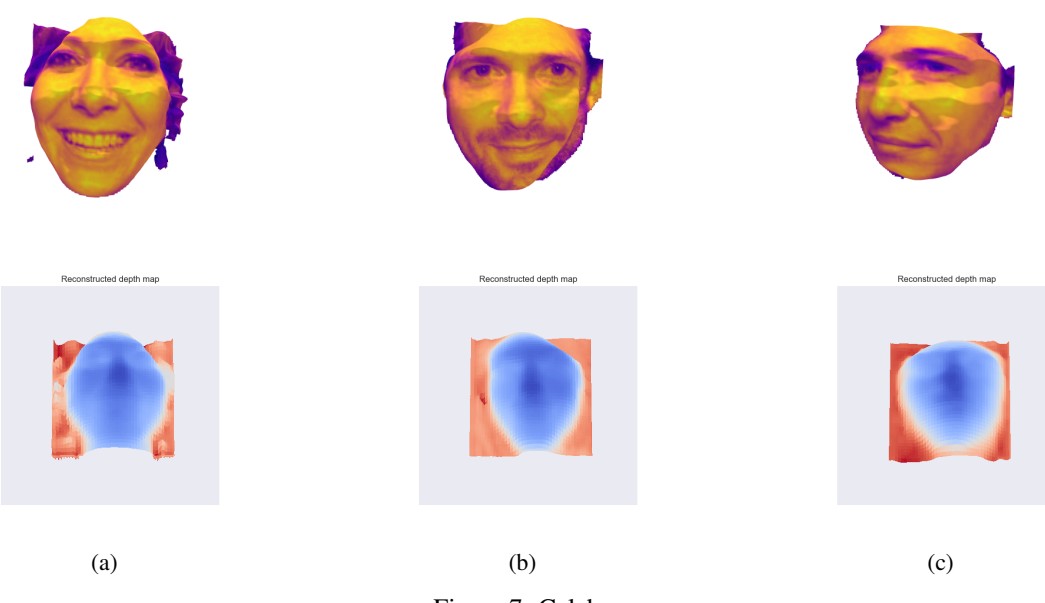

(a)  (b)  (c)

Figure 7: Celeba

## A.3  Effects of shape priors

Figure 8 shows the effects of random initialization of the depth network. Figure 9 shows the results on the first car

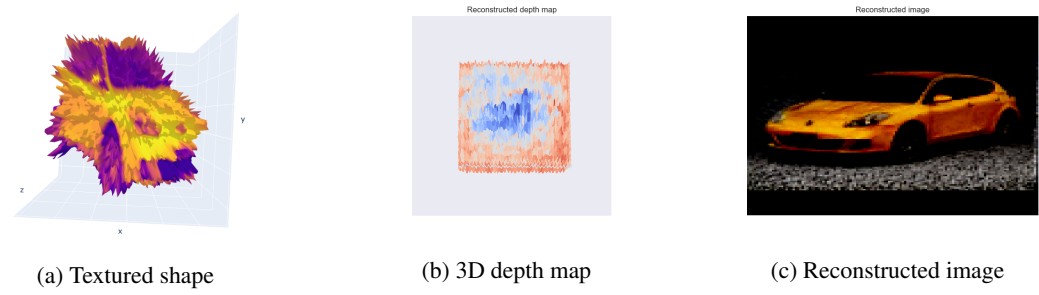

(a) Textured shape  (b) 3D depth map  (c) Reconstructed image

Figure 8: Results with no shape prior.

where it can be observed that our prior is even better the the ellipsoid at capturing fine details such as the side mirror.

**Confidence-Based Prior.** Another experiment we performed focused on the performance of the second prior we presented, the confidence based prior. Figure 10 shows some results on the datasets considered in this paper. The results are most promising in the Celeba dataset where the image of a face is correctly projected even if viewed from the side.

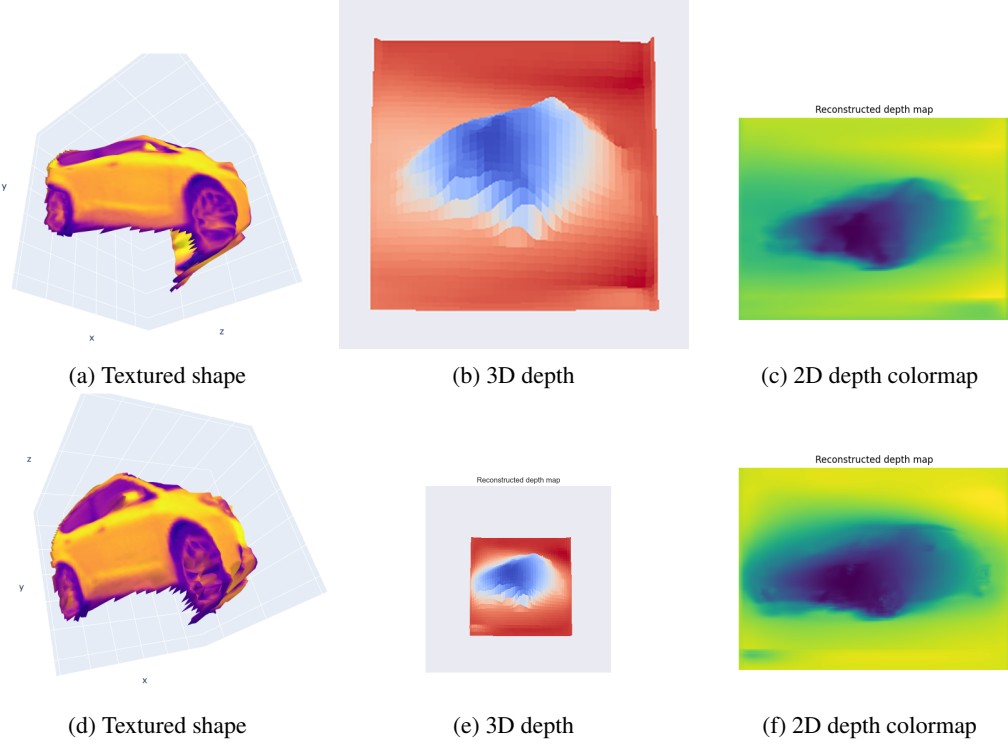

Figure 9: Ellipsoid prior (top row) vs. the **smoothed masked box** (bottom row) prior.

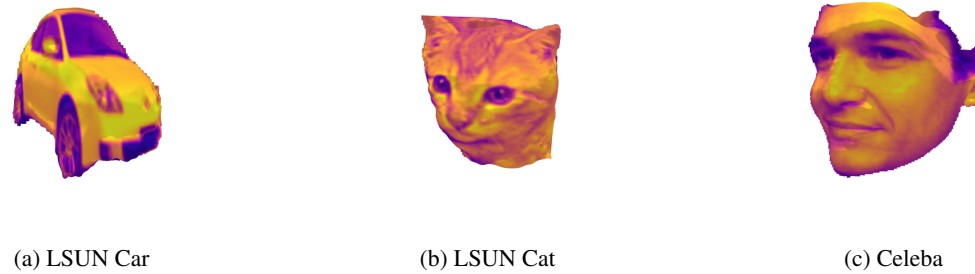

Figure 10: Results with the confidence based prior.

### A.4   Variability of identical runs

### A.5   Catastrophic forgetting

When the training process is complete for one image $\mathbf{I}_t$ we have confirmed that the model $M_t = \{V, L, D, A\}_t$ is able to construct a believable depth map (subsection 4.1). However, when training continues to the next image $\mathbf{I}_{t+1}$ and $M_{t+1}$ is obtained, we have observed that the ability to predict the depth map of the previous image deteriorates, and the problem gets worse with an increasing time discrepancy between the model and image. In other words, the depth network $D_t$ at training step $t$ is only usable for predicting the depth map $\mathbf{d}_t = D_t(\mathbf{I}_t)$ and so suffers from catastrophic forgetting of the previous images. This is illustrated in Figure 13.

The training time for one $128 \times 128$ pixel RGB image using a Tesla K80 GPU was about 2.5 hours, which seems exceedingly costly for just one low-resolution depth map.

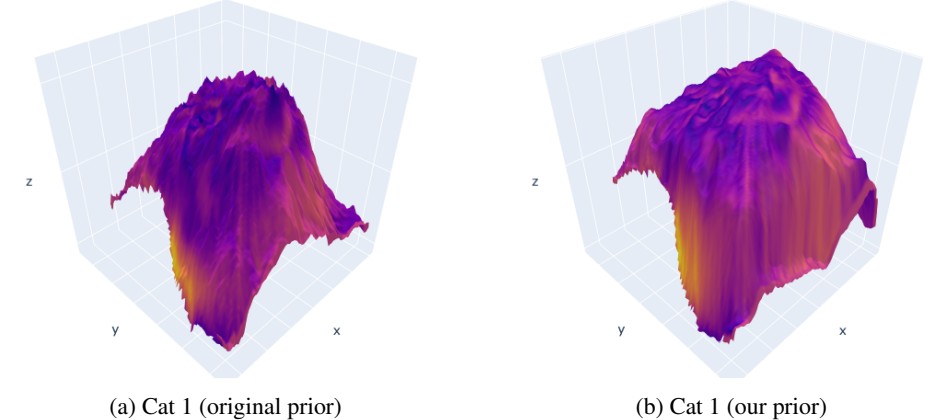

(a) Cat 1 (original prior)         (b) Cat 1 (our prior)

Figure 11: Results for a few other images from the LSUN Cat dataset, for the ellipsoid (left) and smoothed masked box (right) priors.

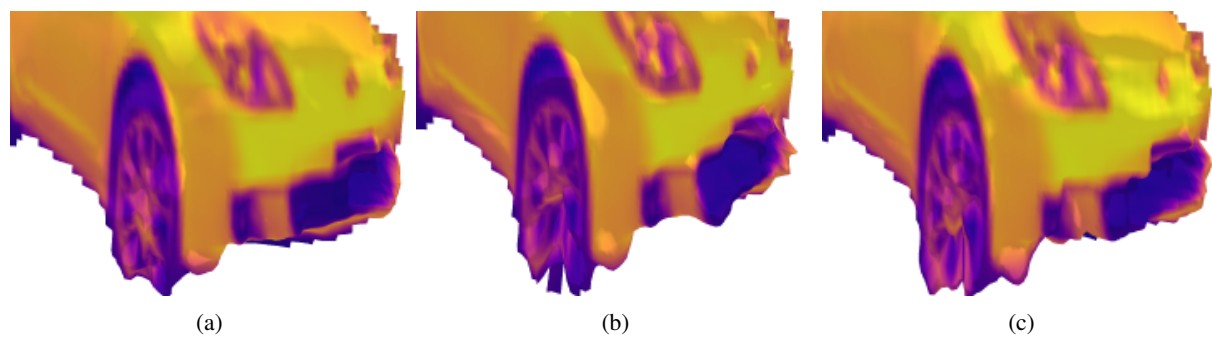

(a)           (b)           (c)

Figure 12: Several runs with identical configuration.

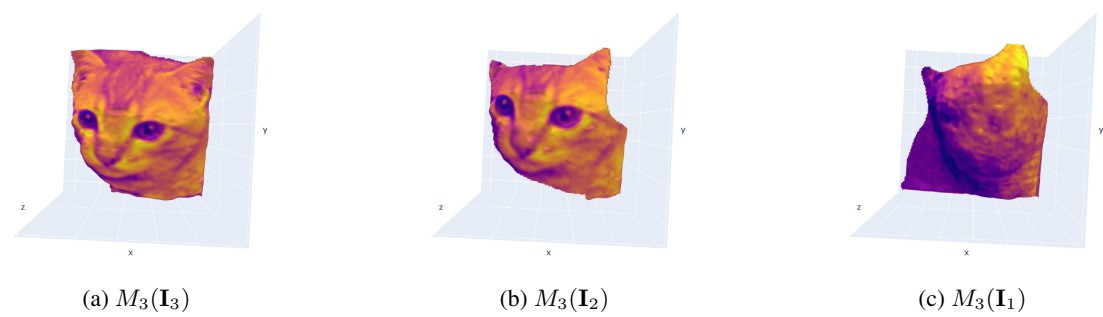

(a) $M_3(\mathbf{I}_3)$      (b) $M_3(\mathbf{I}_2)$      (c) $M_3(\mathbf{I}_1)$

Figure 13: Depth map predictions for a few image samples from the LSUN Car dataset, illustrating catastrophic forgetting for the model $M$.

## A.6 Additional generalized training results

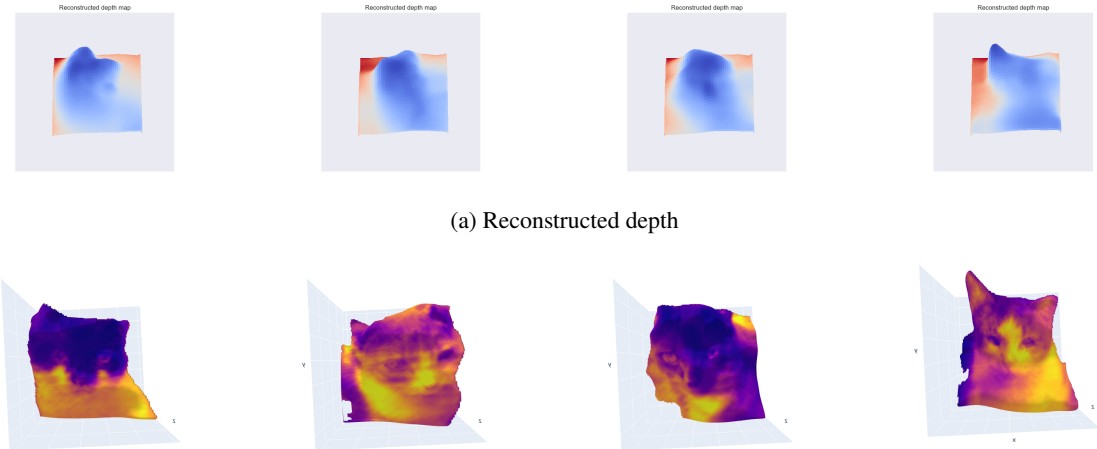

(a) Reconstructed depth

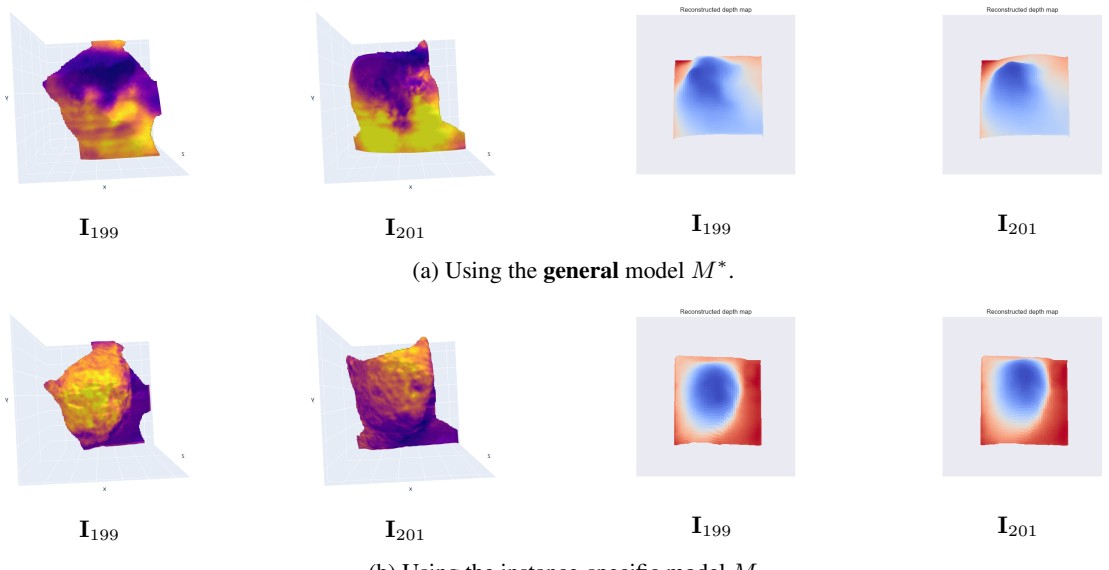

(b) Reconstruced 3D image

Figure 14: Depth map predictions for a few image samples from the training set $\mathcal{D} \subset$ LSUN Cat dataset, all using one and the same **general** model $M^*$.

$\mathbf{I}_{199}$   $\mathbf{I}_{201}$   $\mathbf{I}_{199}$   $\mathbf{I}_{201}$

(a) Using the **general** model $M^*$.

$\mathbf{I}_{199}$   $\mathbf{I}_{201}$   $\mathbf{I}_{199}$   $\mathbf{I}_{201}$

(b) Using the instance-specific model $M_{\text{last}}$

Figure 15: Depth map predictions for **unseen** image samples $\{\mathbf{I}_{199}, \mathbf{I}_{201}\} \notin \mathcal{D}$ from the LSUN Cat dataset.

## A.7   Initialization iteration results

The observations of sections 5.3.1 and 3.4 can be condensed into two main points to form a hypothesis. Please note that our limited computational resources meant that we could not perform rigorous experimentation to confirm these observations with a large number of samples, and that this section should be viewed as a speculative experiment.

- The initial few training iterations can be viewed as an *initialization* of the weights, which depends on what projected samples are generated by the StyleGAN2 model.

- The "features" (i.e. peaks and valleys) of the depth map predictions do not qualitatively change with increasing iterations, but remain fixed except in size (i.e. the height of the peaks).

If one accepts these claims, then it is clear that the first few iterations determine the success of the shape reconstruction. That is why we experiment with drastically increasing the number of pseudo-samples during the first iterations. This

reduces the bias of the initialization and reduces the relative impact that a poor projected sample generated by the GAN has on the model weights. Specifically, we increase the number of projected samples $N_p$ from 16 to 128 for 10 short epochs, in which each training step is performed only once.

Ideally, one would of course permanently increase $N_p$, but with extreme costs in terms of training time. This method only added $\sim 4$ minutes of training time using a Tesla T4 GPU.

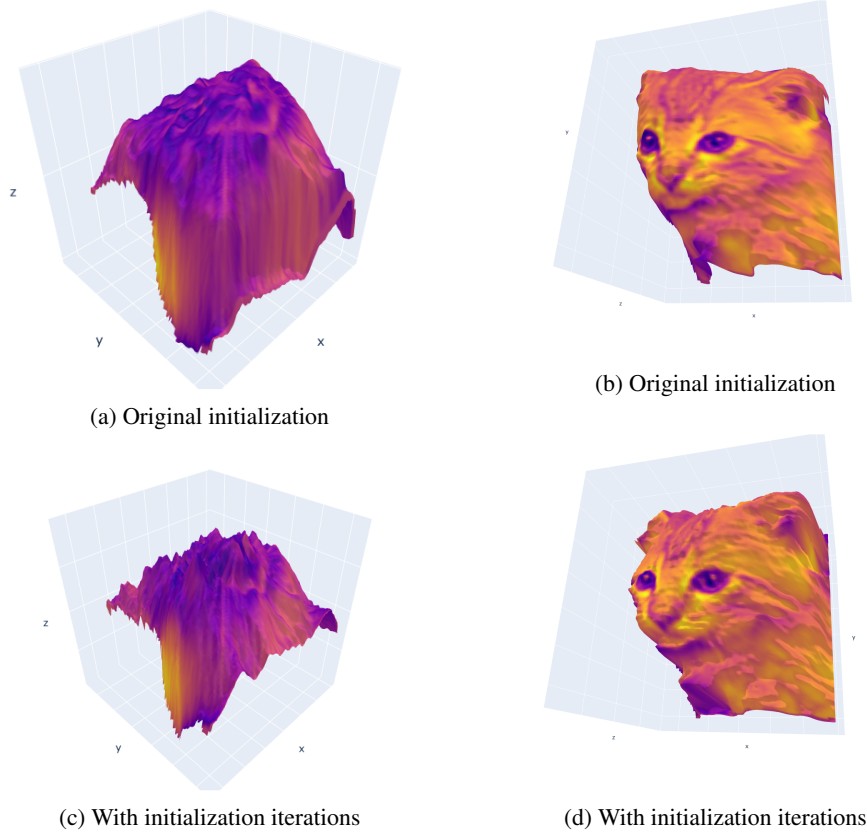

(a) Original initialization

(b) Original initialization

(c) With initialization iterations

(d) With initialization iterations

Figure 16: Results for the worst performers for the **single-image** model using the smoothed box prior, from the LSUN Cat dataset. Original initialization (top row) and using **initialization iterations** (bottom row). The leftmost cat saw the most drastic changes. While the result is a "spikey" depth map, we argue that the general shape has a better resemblance to a cat, and less to a square box-like in the original initialization. The rightmost cat saw some improvement in some details such as the ears and the mouth region.

## A.8  Hyperparameter tuning

We found that $N_p$ correlates with the quality of the predicted shapes. The trend tends to be that more is better, but with diminishing returns. The biggest benefit that a large $N_p$ has, is that strange artefacts are less likely to persist. It is difficult to pinpoint an acceptable threshold for $N_p$, as it varies between datasets and even between images. Therefore we believe a good compromise is to perform a few initialization iterations as described in section 4.2.3 with a large $N_p$ (i.e. 128) and then continue training with a lower number according to the aforementioned thresholds.

To illustrate the results when varying on the number of projected samples $N_{proj}$ we present the results on the LSUN car and Celeba dataset. In Figure 18 the first two cars (corresponding to a low $N_p$) have more irregular surfaces and one has a large spike, while the third is more regular. The same is observed for the Celeba faces in Figure 17, where the first face (corresponding to a low $N_p$) has significant irregularities across the face. As described in subsubsection 5.3.1, we attribute this phenomenon to lower relative impact that sampling poor view-light projections has, the larger $N_p$ is.

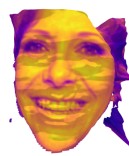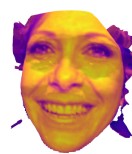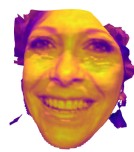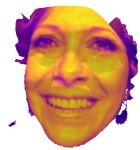

Figure 17: Face 1 when trained with 4, 8, 16 and 32 (from left to right) number of projected samples.

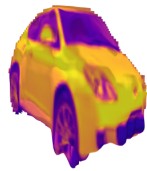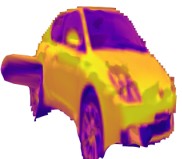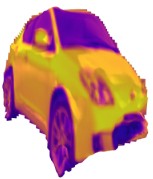

Figure 18: Car 1 when trained with 2, 4 and 8 (from left to right) number of projected samples.

