# OpenReview forum: "Replicating and Improving GAN2Shape Through Novel Shape Priors and Training Steps"
_ML_Reproducibility_Challenge/2021/Fall — RC2021_

### Official Review · Reviewer_XqhS · 2022-02-20
**GAN2Shape for 3D shape recovery and identify its strengths and weaknesses.**

**Rating:** 10
**Confidence:** 5

**Review:**

The study objective is interesting. The manuscript is written very nicely. The authors shared a link, which contains codes. The results are a significant improvement. The work is reproducible.

---

### Official Review · Reviewer_SXvA · 2022-03-09
**Good reproducibility study, writing and structure needs to be improved**

**Rating:** 6
**Confidence:** 4

**Review:**

This paper aims to reproduce "Do 2D GANs Know 3D Shape? Unsupervised 3D Shape Reconstruction from 2D Image GANs" by Pan et al.

The authors provide a complete and clear summary of the work that has been done. They also state clearly which of the research claims made by Pan et al. are subject to this reproducibility study. The authors decide to re-implement the entire code-base, relying on external dependencies where necessary. Their initial attempt to write the code from scratch relying only on the documentation in the original paper had to be abandoned due to lack of details. The code of the authors seems well enough documented with clear instructions on how to run it. The authors try out additional hyperparameters that have not been reported by the original authors. They additionally experiment with the number of projected samples and the number of epochs for the pre-training.

The authors did not communicate with the original authors. Clarifying the different implementation of the loss function compared to the paper would have been helpful.

Despite the effort of reproducing the authors also go beyond the original work and try out different shape priors and introduce a generalized training procedure. The motivation for the novel shape priors is missing. Nonetheless, this additional experiment is helpful in understanding the original work. Their newly proposed 3D shape priors seem to improve shape reconstruction for some cases. The proposal for a generalized training structure seems reasonable to improve generalization and is backed up with their experiments on catastrophic forgetting. The improvement achieved by their improved initialization seems debatable.

The authors clearly discuss which parts of the original paper could be reproduced and which could not.

The paper provides useful experiments to validate the research claims by Pan et al. They carry out many additional experiments which are useful. The structure of the paper is confusing. I urge the authors to improve the writing and especially the structure of the paper, avoiding heavy cross-referencing between results, earlier sections, appendix, and conclusions which makes it very hard to follow.


General remarks:

- The paper contains many small experiments that would benefit from being better connected.
- The structure of the paper could be improved by first presenting their experiments, results, and conclusions of the reproduction and in the following section discuss own experiments, results, and conclusion. This avoids references to findings later in the paper (see for example line 112). Generally, references to appendices and different sections are hard to follow and are very confusing to the reader.
- Reference to 3.1.1. in line 173 does not contain the hypothesis “…that this method would not work at all without a shape prior…”.
- When referring to the appendix the exact subsection should be mentioned, otherwise, it is very hard to follow what the authors refer to, e.g. line 178
- Generally, when referring to observations in different sections it would be helpful to repeat them briefly, e.g. see section 4.2.3 line 189.
- Appendix 4 does not seem not to contain any content.
- The Confidence-Based Prior results are not referenced in the paper and can be only found in the appendix
- When claiming superior results it would be helpful to have reference images text to the improved ones to compare with directly. Figure 10.

Grammatical issues:

- use either pre-training or pertaining
- we observe see marginal -> we observe marginal, line 191
- and larger improves in -> and larger improvements in, line 191

---

### Meta-Review · Program_Chairs · 2022-04-09

**Recommendation:** Accept
**Confidence:** 5

**Metareview:**

A solid contribution to the reproducibility challenge.  The submission has been accepted.

---

### Decision · Program_Chairs · 2022-04-09

**Decision:**

Accept

**Comment:**

Following the recommendation of reviewers and meta-reviewer, the paper is accepted for ML Reproducibility Challenge 2021, and will be published in the upcoming special edition of ReScience Journal.